# Amphiphilic Nucleobase-Containing Polypeptide Copolymers—Synthesis and Self-Assembly

**DOI:** 10.3390/polym12061357

**Published:** 2020-06-16

**Authors:** Michel Nguyen, Khalid Ferji, Sébastien Lecommandoux, Colin Bonduelle

**Affiliations:** 1Laboratoire de Chimie de Coordination, UPR CNRS 8241, 205 route de Narbonne, CEDEX 04, 31077 Toulouse, France; michel.nguyen@lcc-toulouse.fr; 2University of Lorraine, CNRS, LCPM, F-54000 Nancy, France; khalid.ferji@univ-lorraine.fr; 3University of Bordeaux, CNRS, Bordeaux INP, LCPO, UMR 5629, F-33600 Pessac, France; lecommandoux@enscbp.fr

**Keywords:** sequential ring-opening polymerization, amphiphilic polypeptide, nucleobase, spontaneous self-assembly, DNA binding

## Abstract

Nucleobase-containing polymers are an emerging class of building blocks for the self-assembly of nanoobjects with promising applications in nanomedicine and biology. Here we present a macromolecular engineering approach to design nucleobase-containing polypeptide polymers incorporating thymine that further self-assemble in nanomaterials. Diblock and triblock copolypeptide polymers were prepared using sequential ring-opening polymerization of γ-Benzyl-l-glutamate *N*-carboxyanhydride (BLG-NCA) and γ-Propargyl-l-glutamate *N*-carboxyanhydride (PLG-NCA), followed by an efficient copper(I)-catalyzed azide alkyne cycloaddition (CuAAc) functionalization with thymidine monophosphate. Resulting amphiphilic copolymers were able to spontaneously form nanoobjects in aqueous solutions avoiding a pre-solubilization step with an organic solvent. Upon self-assembly, light scattering measurements and transmission electron microscopy (TEM) revealed the impact of the architecture (diblock versus triblock) on the morphology of the resulted nanoassemblies. Interestingly, the nucleobase-containing nanoobjects displayed free thymine units in the shell that were found available for further DNA-binding.

## 1. Introduction

Amphiphilic block copolymers are known to self-assemble in aqueous solutions into a wide range of nanostructures including worm-like or spherical micelles and vesicles [1,2,3]. In this field, polypeptide-based copolymers show considerable promise as building blocks to design bioinspired nanomaterials interfacing living systems [4,5]. Compared to other synthetic copolymers, those made of amino acids may offer several attractive features, including (1) adoption of ordered secondary conformations (α-helices, β-sheets…) as in natural proteins; (2) improvement of chemical and enzymatic degradability in a biological medium; and (3) providing tailorable and metabolizable building units/blocks [6,7]. In addition, the supramolecular organization of polypeptide offers a unique opportunity to produce multivalent structures and can also be used to promote selective interactions with biological targets [8,9,10].

Varying the secondary structure of polypeptides is a unique chemical leverage in polymer science [6,11]. When secondary structure is included in amphiphilic macromolecular design, the polypeptide segment often provides rigidity and promotes the formation of lamellar structures [12]. For instance, amphiphilic polypeptides with α-helical “rod” hydrophobic block have been extensively used to prepare vesicles [13,14]. In this direction, poly(γ-benzyl-l-glutamate) (PBLG) has a strong tendency to adopt α-helical conformation—it can be used to design amphiphilic copolymers whose self-assembly results in bioactive nanomaterials, [15]. Among those nanomaterials, PBLG afforded amphiphilic copolymers that “spontaneously” self-assemble in aqueous solution when combined with hydrophilic segment exhibiting dendritic shape [16]. Herein, we show that combining PBLG segments with polypeptides containing charged nucleobases also promote such “spontaneous” self-assembly.

Currently, nucleobase-containing polypeptides produced upon a one step polymerization process are rare and synthetic peptide nucleic acid (PNA) oligomers are mostly prepared using multistep coupling methods [17,18]. Block copolymers combining polypeptide and nucleic acid have previously been used for the design of molecular bottlebrush giving rise to unprecedented hydrogels [19,20]. In addition, direct grafting of nucleobases onto simple poly(γ-propargyl-l-glutamate) (PPLG) scaffold have afforded unique structuring switch upon DNA binding [21] or ionic polypeptides with unusual beta sheet stability [22]. Herein, diblock and triblock copolypeptides, PBLG-*b*-PPLG and PPLG-*b*-PBLG-*b*-PPLG respectively, were prepared using sequential ring-opening polymerization of γ-Benzyl-l-glutamate *N*-carboxyanhydride (BLG-NCA) and γ-Propargyl-l-glutamate *N*-carboxyanhydride (PLG-NCA). The resulting copolypeptides were functionalized by thymidine monophosphate using copper(I)-catalyzed azide alkyne cycloaddition (CuAAC) to prepare the targeted nucleobase-containing polypeptide copolymers. Self-assembly of such copolypeptides was induced spontaneously by a direct solubilization in water. The morphology of nanoobjects formed were characterized using light scattering and transmission electron microcopy. Through this study, we showed that nucleobase-containing polypeptides paved the way to unprecedented amphiphilic copolymers fully based on amino acid and nucleotide building blocks. Their simple self-assembly in aqueous solution resulted in nanomaterials whose hydrophilic shells also promoted efficient DNA binding.

## 2. Materials and Methods

**Materials and instruments.** Dimethyl formamide (DMF), tetrahydrofuran (THF) and dichloromethane (CH_2_Cl_2_) (Sigma Aldrich, EU) were purified through a solvent purification system and systematically purged with dry argon before use. Hexylamine and ethylene diamine initiators (Sigma-Aldrich, EU) were distilled before use. l-glutamic acid, propargyl alcohol, chlorotrimethylsilane, diethyl ether, diphosgene, *N*,*N*,*N*′,*N*″,*N*″-Pentamethyl diethylenetriamine (PMDETA) and copper bromide (CuBr) were used as received (Sigma Aldrich, EU). γ-Benzyl-l-glutamate *N*-carboxyanhydride (BLG-NCA) was used as received (PMC Isochem, France). 3′-azido-2′, 3′-dideoxythymidine monophosphate (AZTP) was used as received (Jena Bioscience, Germany). The DNA oligomers were purchased from Genscript (EU).

^1^H NMR (Nuclear Magnetic Resonance) spectra were recorded on a Bruker AC 400 spectrometer. The reported chemical shifts are relative to CDCl_3_, DMF-*d^7^* or D_2_O. Infrared spectroscopy was monitored on a Bruker Tensor 27 spectrometer using the attenuated total reflection (ATR) method. Polymer molar masses were determined by size-exclusion chromatography (SEC) using DMF/LiBr (1 g/L) as the eluent at 55 °C. Measurements were performed on an Ultimate 3000 system from ThermoScientific equipped with diode array detector (DAD). The system also includes a multi-angles light scattering detector MALS and differential refractive index detector dRI from Wyatt technology. Polymers (5 mg/mL) were separated on three Shodex Asahipack gel columns [GF-1G 7B (7.5 × 8 mm), GF 310 (7.5 × 300 mm), GF510 (7.5 × 300), exclusion limits from 500–300,000 Da] at a flowrate of 0.5 mL/min. Easivial kit of Polystyrene from Agilent was used as a standard (*M*_n_ from 162 to 364,000 Da). Transmission Electron Microscopy (TEM) images were recorded on a JEOL USA Transmission Electron Microscopes working at 100 kV. Samples were prepared by spraying a 1 g/L solution of the block copolymer onto a copper grid (200 mesh coated with carbon) using a homemade spray tool and negatively stained with 1.2% uranyl acetate aqueous solution.

Light Scattering measurements were performed using an ALV/CGS-3 compact goniometer system, equipped with an ALV-7004 multiple tau digital correlator and a vertically-polarized He-Ne laser of 22 mW output power operating at wavelength λ = 632.8 nm at 20 °C. Scattered light was detected at scattering angles 20° ≤ θ ≤ 150° corresponding to scattering vector regime $0.00459{\;\mathrm{nm}}^{-1}\;\leq \;q\;\leq \;0.00255{\;\mathrm{nm}}^{-1}$. Z-average hydrodynamic radius (R_H_) was calculated using the Stokes-Einstein equation (Equation (1)), where D_0_ is the coefficient diffusion of the nanoobjects determined from the slope of the q^2^ dependence of relaxation rate (<Γ> = Dq^2^), k_B_ is the Boltzmann constant, T is the absolute temperature and *η_s_* is the viscosity of the solvent (water). The Kc/R_θ_ ratio was recorded in order to determine the radius of gyration (R_g_) using the Guinier approximation (Equation (2)), where R_θ_ is the average scattered intensity measured at angle θ, K is an optical constant, c is the concentration of the nanoobject dispersion and M_W,NPs_ is the weight-average molar mass of the nanoobjects [23]. The R_g_ is obtained from the slope of the straight line of the plot $\ln\left(\frac{Kc}{R_{\theta }}\right)\mathrm{vs}\;q2$ (where qR_g_ < 1).
(1)$$R_{H}=\frac{k_{B}T}{6\pi \eta_{s}D_{0}}$$
(2)$$\ln\left(\frac{Kc}{R_{\theta }}\right)\approx \frac{1}{3}R_{g}^{2}q^{2}-\;\ln\left(M_{W,NPs}\right)$$

**Synthesis of γ-Propargyl-*l*-glutamate *N*-carboxyanhydride (PLG-NCA)** was prepared as reported by us with some modifications [21].

1)*Synthesis of γ-propargyl l-glutamate*. l-glutamic acid (10 g, 68 mmol), propargyl alcohol (370 mL) and chlorotrimethylsilane (19 mL, 150 mmol) were stirred together at 0 °C (dropwise addition of the silane) for 1 h and then left stirring at room temperature (RT) for two days. Upon filtration, precipitation into diethyl ether twice to yield 9.98 g of a white solid (66%). ^1^H NMR (400 MHz, D_2_O, δ, ppm) 2.19 (m, 2H, CO-CH_2_-C**H_2_**-CH), 2.61 (dt, 2H, CO-C**H_2_**-CH_2_-CH), 2.85 (t, 1H, C≡CH), 4.02 (t, 1H, CH), 4.68 (d, 2H, CH_2_O) ^13^C NMR (100.6 MHz, D_2_O, δ, ppm) 24.74 (CO-**C**H_2_-CH_2_-CH), 29.42 (CO-CH_2_-**C**H_2_-CH), 52.14 (CH), 52.77 (CH_2_O), 76.07 (C≡**C**H), 77.60 (**C**≡CH), 171.67 (**C**O-CH_2_), 173.44 (**C**(O)OH). Fourrier-transformed Infrared Spectroscopy (FTIR) 1747 cm^−1^, 1725 cm^−1^. High Resolution Mass Spectrometry (HRMS, ESI, positive, m/z) calculated for C_8_H_12_NO_4_, 186.0766, found 186.0766.2)*Synthesis of N-carboxyanhydride of γ-propargyl l-glutamate.* γ-propargyl l-glutamate hydrochloride (5 g, 22.56 mmol) was stirred in THF at room temperature (RT). Careful addition of diphosgene (*highly toxic*, 3.11 g, 15.74 mmol) was performed and the reaction was left stirring for 2 h. The solvent was then fully removed and the solid residue washed many times with pentane. The NCA was ultimately dissolved in a minimal amount of dry CH_2_Cl_2_ and passed through a column of dry silica gel [24] to afford a yellowish powder (4.7 g, 99%). ^1^H NMR (400 MHz, CDCl_3_, δ, ppm) 2.25 (m, 2H, COCH_2_C**H_2_**CH), 2.54 (t, 1H, C≡CH), 2.63 (t, 2H, COC**H_2_**CH_2_CH), 4.45 (t, 1H, CH), 4.73 (d, 2H, CH_2_O), 6.77 (s, 1H, NH). ^13^C NMR (100.6 MHz, D_2_O, δ, ppm) 26.78 (CO-CH_2_-**C**H_2_-CH), 29.46 (CO-**C**H_2_-CH_2_-CH), 52.63 (CH_2_O), 56.79 (CH), 75.52 (C≡**C**H), 77.05 (**C**≡CH), 151.95 (**C**O-NH), 169.31 (**C**(O)O), 171.71 (**C**(O)CH_2_).FTIR 1854 cm^−1^, 1782 cm^−1^, 1655 cm^−1^.


**Synthesis of amphiphilic diblock nucleopolypeptides.**


1)*Synthesis of poly(γ-benzyl-l-glutamate), PBLG, monofunctional macroinitiator initiated by hexylamine.* The *N*-carboxyanhydride (NCA) monomer of γ-benzyl-*l*-glutamate (2 g, 7.6 mmol) was dissolved in 10 mL of dry DMF in a Schlenk tube. A solution of hexylamine (50 µL, 0.38 mmol) in 2 mL dry DMF was added and the reaction was left to stir in a cold bath at 0 °C for 4 days under argon. After BLG-NCA had been completely consumed as monitored by FTIR (disappearance of the peaks at 1787 cm^−1^ and 1854 cm^−1,^ see Appendix A) and NMR, an aliquot of the macroinitiator was taken (0.79 mL, 90 mg after Et_2_O precipitation) under argon, the left solution was used for synthesis of the diblock as described below. ^1^H NMR (400 MHz, CDCl_3_/TFA 15%, δ, ppm) 0.89 (t, CH_3_ hexylamine), 1.28 (m, CH_3_C**H_2_**C**H_2_**C**H_2_**CH_2_CH_2_ hexylamine), 1.49 (m, CH_3_CH_2_CH_2_CH_2_C**H_2_**CH_2_ hexylamine), 2.03 (m, 2H, COCH_2_C**H_2_**CH), 2.46 (m, 2H, COC**H_2_**CH_2_CH), 3.24 (m, CH_3_CH_2_CH_2_CH_2_CH_2_C**H_2_** hexylamine), 4.61 (m, 1H, CH), 5.10 (m, 2H, CH_2_O), 7.31 (m, 5H, Ph), 7.86 (m, 1H, NH).^13^C NMR (100.6 MHz, CDCl_3_/TFA 15%, δ, ppm) 13.47 (CH_3_ hexylamine), 22.23 (CH_3_**C**H_2_ hexylamine), 26.15 (CH_3_CH_2_**C**H_2_ hexylamine), 26.79 (CO-CH_2_-**C**H_2_-CH), 28.33 (CH_3_CH_2_CH_2_**C**H_2_ hexylamine), 29.86 (CO-**C**H_2_-CH_2_-CH), 31.05 (CH_3_CH_2_CH_2_CH_2_**C**H_2_ hexylamine), 40.77 (CH_3_CH_2_CH_2_ CH_2_CH_2_**C**H_2_ hexylamine) 53.21 (CH), 68.16 (CH_2_O), 128.14 (CHPh), 134.19 (CPh), 173.08 (CO-NH), 175.29 (C(O)OPh). FTIR 1651 cm^−1^ (amide I) 1729 cm^−1^ (ester). DP (^1^H NMR): 19. SEC provided a number-average molecular weight Mn of 2500 g/mol (calibration with polystyrene standards) and a dispersity (Ð = *M*_w_/*M*_n_) of 1.10.2)*Polymerization of γ-propargyl-l-glutamate from PBLG macroinitiator—Synthesis of poly(γ-benzyl-l-glutamate–b–γ-propargyl-l-glutamate), PBLG-b-PPLG.* A solution of PLG-NCA (1.83 g, 8.68 mmol) in DMF (3.9 mL) was added to the residual solution of PBLG macroinitiator and the reaction mixture was left stirring at RT for overnight. After full conversion (as monitored by FTIR following the disappearance of the peak at 1787 cm^−1^, see Appendix A), the reaction mixture was precipitated into an excess of cold Et_2_O and recovered after centrifugation as a white solid (2.34 g, 79% yield). ^1^H NMR (400MHz, CDCl_3_/TFA 15%, δ, ppm) 0.88 (t, CH_3_ hexylamine), 1.27 (m, CH_3_C**H_2_**C**H_2_**C**H_2_**CH**_2_**CH**_2_** hexylamine), 1.49 (m, CH_3_CH_2_CH_2_CH_2_C**H_2_**CH**_2_** hexylamine), 1.99 (m, 2H, PBLG, COCH_2_C**H_2_**CH), 2.17 (m, 2H, PPLG, COCH_2_C**H_2_**CH), 2.45 (m, 2H, PBLG, COC**H_2_**CH_2_CH), 2.51 (m, 1H, PPLG, C≡CH), 2.57 (m, 2H, PPLG, COC**H_2_**CH_2_CH), 3.24 (m, CH_3_CH_2_CH_2_CH_2_CH_2_C**H_2_** hexylamine), 4.59 (m, 1H, PBLG, CH), 4.66 (m, 1H, PPLG, CH), 4.72 (m, 2H, PPLG, CH_2_O), 5.09 (m, 2H, PBLG, CH_2_O), 7.30 (m, 5H, Ph), 7.86 (m, 1H, PBLG, NH), 7.94 (m, 1H, PPLG, NH). ^13^C NMR (100.6 MHz, CDCl_3_/TFA 15%, δ, ppm) 26.60 (PPLG, CO-CH_2_-**C**H_2_-CH), 26.71 (PBLG, CO-CH_2_-**C**H_2_-CH), 29.72 (PPLG, CO-**C**H_2_-CH_2_-CH), 29.89 (PBLG, CO-**C**H_2_-CH_2_-CH), 53.28 (PBLG, CH), 53.35 (PPLG, CH), 68.09 (PBLG, CH_2_O), 75.66 (PPLG, CH_2_O), 128.15-128-67 (CHPh), 134.25 (CPh), 173.07 (PBLG, CO-NH), 173.14 (PPLG, CO-NH), 174.28 (PPLG, C(O)O), 175.20 (PBLG, C(O)OPh). DP from ^1^H NMR: 19 (PBLG) +25 (PPLG). SEC then provided a number-average molecular weight M_n_ of 5900 g/mol (calibration with polystyrene standards) and a low dispersity (Ð = 1.13).3)*Functionalization of PBLG-b-PPLG with zidovudine using* CuAAC reaction—*Synthesis of nucleopolypeptide diblock copolymer (PBLG-b-PPLGNu)*. PBLG-*b*-PPLG (63 mg, ca. 0.19 mmol of alkyne units), AZTP (100 mg, 0.39 mmol, 2 equiv. to alkyne groups) and CuBr (8.6 mg, 0.06 mmol, 0.33 equiv. to alkyne groups) were dissolved in 5 mL of anhydrous and deoxygenated DMF in a Schlenk tube. After 3 freeze thawing, PMDETA (12.6 mg, 0.06 mmol, 0.33 equiv. to alkynes groups) was added and the Schlenk tube was placed in an oil bath at 25 °C for 16 h. The crude product was purified by dialysis for 4–5 days against milliQ water (Spectra/Por MWCO 3.5 kDa membrane), containing ethylenediaminetetraacetic acid (EDTA) the first 2 days, acidic conditions for 2 h (pH = 2) and milliQ water the rest of the time. Finally, the polymer was recovered upon lyophilization (yield 94%). ^1^H NMR (400 MHz, D_2_O, δ, ppm) 0.94 (t, C**H_3_** hexylamine), 1.20–1.42 (m, 3 × 2H, CH_3_C**H_2_**C**H_2_**C**H_2_**CH**_2_**CH**_2_** hexylamine), 1.68 (m, 2H, CH_3_CH_2_CH_2_CH_2_C**H_2_**CH**_2_** hexylamine), 2.22 (s, C**H_3_** AZTP), 2.37–2.55 (C**H_2_** ribose AZTP), 2.58–2.92 (m, COCH_2_C**H**HCH and COCH_2_CH**H**CH of the side chains), 3.05–3.19 COC**H_2_**CH_2_CH of the side chains), 4.50 (s broad, C**H_2_**OP AZTP), 4.62 (s broad, COC**H**NH of both monomer units), 4.88 (s broad, C**H** ribose AZT), 5.05–5.45 (m broad, COOC**H_2_**C_6_H_5_), 5.53 (s broad, COOC**H_2_**-triazole), 5.84 (m, 48H, C**H** ribose AZTP), 6.76 (m broad, C**H** ribose AZTP), 7.35 (m broad, CH_2_C_6_**H_5_**), 8.14 (s broad, C**H** thymidine AZTP), 8.52 (s broad, 49H, C**H** triazole). ^31^P NMR (162 MHz, D_2_O, δ in ppm): −0.05 ppm (broad). FTIR—disappearance of the N_3_ stretching at 2109 cm^−1^ and broad phosphate stretching at 1695 cm^−1^. Amide I 1649 cm^−1^ and amide II 1550 cm^−1^ (Alpha helix stretching comes from PBLG).


**Synthesis of amphiphilic triblock nucleopolypeptide**


1)*Synthesis of poly(γ-benzyl-l-glutamate), PBLG-ED-PBLG, difunctional macroinitiator initiated by ethylenediamine.* The NCA monomer of γ-benzyl-l-glutamate (2 g, 7.6 mmol) was dissolved in 10 mL dry DMF in a Schlenk tube. A solution of ethylenediamine (12.7 µL, 0.19 mmol) in 2 mL dry DMF was added and the reaction was left to stir in a cold bath at 0 °C for 4 days under inert atmosphere. After BLG-NCA had been completely consumed as monitored by FTIR (disappearance of the peaks at 1787 cm^−1^ and 1854 cm^−1^, see Appendix A) and NMR, an aliquot of the macroinitiator was taken (0.79 mL, 0.11 g after Et_2_O precipitation). 3 aliquots of 70 µL were used to follow the polymerization reaction by FTIR. The solution was used for synthesis of the triblock (cf. below). ^1^H NMR (400MHz, CDCl_3_/TFA 15%, δ, ppm) 2.03 (m, 2X2H, COCH_2_C**H_2_**CH), 2.46 (m, 2X2H, COC**H_2_**CH_2_CH), 4.61 (m, 2X1H, CH), 5.10 (m, 2X2H, CH_2_O), 7.31 (m, 2X5H, Ph), 7.86 (m, 2X1H, NH). ^13^C NMR (100.6 MHz, CDCl_3_/TFA 15%, δ, ppm) 26.79 (CO-CH_2_-**C**H_2_-CH), 29.86 (CO-**C**H_2_-CH_2_-CH), 53.21 (CH), 68.16 (CH_2_O), 128.14 (CHPh), 134.19 (CPh), 173.08 (CO-NH), 175.29 (C(O)OPh). [^1^H NMR Initiator: 3.40 (m, 2X2H, CH_2_ ethylenediamine)]. FTIR 1652-1646 cm^−1^ (amide I), 1729 cm^−1^ (ester). DP (^1^HNMR): 19 (X2). SEC then provided a number-average molecular weight Mn of 5000 g/mol (calibration with polystyrene standards) and a low dispersity (Ð = 1.08).2)*Polymerization of γ-propargyl-l-glutamate from PBLG-ED-PBLG macroinitiator—Synthesis of poly(γ-propargyl-l-glutamate-b-γ-benzyl-l-glutamate–b–γ-propargyl-l-glutamate), PPLG-b-PBLG-b-PPLG*. A solution of PLG-NCA (1.84 g, 8.7 mmol) in DMF (3.91 mL) was added to the residual solution of PBLG-ED-PBLG macroinitiator and the reaction mixture was left stirring at RT for another day. After full conversion (as monitored by FTIR following the disappearance of the peak at 1787 cm^−1^, see Appendix A), the reaction mixture was precipitated into an excess of cold diethylether and recovered after centrifugation as a white solid (2.62 g, 88% yield). ^1^H NMR (400MHz, CDCl_3_/TFA 15%, δ, ppm) 2.01 (m, 2X2H, PBLG, COCH_2_C**H_2_**CH), 2.13 (m, 2X2H, PPLG, COCH_2_C**H_2_**CH), 2.46 (m, 2X2H, PBLG, COC**H_2_**CH_2_CH), 2.51 (m, 2X1H, PPLG, C≡CH), 2.58 (m, 2X2H, PPLG, COC**H_2_**CH_2_CH), 4.61 (m, 2X1H, PBLG, CH), 4.67 (m, 2X1H, PPLG, CH), 4.72 (m, 2X2H, PPLG, CH_2_O), 5.10 (m, 2X2H, PBLG, CH_2_O), 7.30 (m, 2X5H, Ph), 7.86 (m, 2X1H, PBLG, NH), 7.94 (m, 2X1H, PPLG, NH). ^13^C NMR (100.6 MHz, CDCl_3_/TFA 15%, δ, ppm) 26.62 (PPLG, CO-CH_2_-**C**H_2_-CH), 26.78 (PBLG, CO-CH_2_-**C**H_2_-CH), 29.68 (PPLG, CO-**C**H_2_-CH_2_-CH), 29.86 (PBLG, CO-**C**H_2_-CH_2_-CH), 53.22 (PBLG, CH), 53.40 (PPLG, CH), 68.15 (PBLG, CH_2_O), 75.64 (PPLG, CH_2_O), 128.14 (CHPh), 134.19 (CPh), 173.08 (PBLG, CO-NH), 174.31 (PPLG, C(O)O), 174.41 (PPLG, CO-NH), 175.29 (PBLG, C(O)OPh). [^1^H NMR Initiator: 3.40 (m, 2X2H, CH_2_ ethylenediamine)]. DP (^1^H NMR): 19X2 (PBLG) + 24X2 (PPLG).3)*Functionalization of PPLG-b-PBLG-b-PPLG with zidovudine using* CuAAC reaction—*Synthesis of amphiphilic triblock nucleopolypeptide*. (PPLGNu-*b*-PBLG-*b*-PPLGNu). PPLG-*b*-PBLG-*b*-PPLG (73 mg, ca. 0.19 mmol of alkyne units), AZTP (100 mg, 0.39 mmol, 2 equiv. to alkyne groups) and CuBr (8.6 mg, 0.06 mmol, 0.33 equiv. to alkyne groups) were dissolved in 5 mL of anhydrous and deoxygenated DMF in a Schlenk tube. After 3 freeze thawing, PMDETA (12.6 mg, 0.06 mmol, 0.33 equiv. to alkynes groups) was then added and the Schlenk tube was placed in an oil bath at 25 °C for 16 h. The crude product was purified by dialysis for 4–5 days against milliQ water (Spectra/Por MWCO 3.5 kDa membrane), containing EDTA the first 2 days and then the polymer was recovered upon centrifugation (yield 91%). ^1^H NMR (400MHz, D_2_O, δ in ppm) 2.11–2.31 (s broad, C**H_3_** AZTP), 2.32–2.58 (C**H_2_** ribose AZTP), 2.69–2.99 (m, COCH_2_C**H**HCH and COCH_2_CH**H**CH of the side chains), 3.04–3.21 COC**H_2_**CH_2_CH of the side chains), 4.49 (s broad, C**H_2_**OP AZTP), 4.62 (s broad, COC**H**NH of both monomer units), 4.87 (s broad, C**H** ribose AZT), 5.05–5.45 (m broad, COOC**H_2_**C_6_H_5_), 5.52 (s broad, COOC**H_2_**-triazole), 5.83 (m, 48H, C**H** ribose AZTP), 6.76 (m broad, C**H** ribose AZTP), 7.35 (m broad, CH_2_C_6_**H_5_**), 8.14 (s broad, C**H** thymidine AZTP), 8.52 (s broad, 49H, C**H** triazole).^31^P NMR (162MHz, D_2_O, δ in ppm): −0.06 ppm (broad). FTIR—disappearance of the N_3_ stretching at 2109 cm^−1^ and broad phosphate stretching at 1695 cm^−1^. Amide I 1649 cm^−1^ and amide II 1550 cm^−1^ (α-helix stretching comes from PBLG).


**Self-assembly of copolypeptides**


Diblock and triblock copolypeptides were dispersed in water at 2.5 mg/mL to form nanoobjects using a direct dispersion method. Self-assembly was further probed by dynamic light scattering and uranyl acetate stained TEM. Overall, amphiphilic structures allowed the preparation of nano-assemblies having rather low size dispersity considering the direct hydration process and the use of polymeric materials.


**DNA binding experiment**


Upon self-assembly, the interaction of oligoDNA with nucleopolypeptide was evidenced by DNA binding experiments. (AAAAA)_4_ oligoDNA (15 µM) was incubated with aqueous solutions of the amphiphilic copolymers (50 µg/mL) in milliQ water at RT. The solution was passed through a VIVASPIN centrifugal concentrator cartridge (MW cut-off 30 KDa) (from Fisher Scientific) until 5 times the volume was filtered out. The Oligo DNA retained was evaluated from UV analyses performed at 260 nm (UV absorbance of Adenine).

## 3. Results and Discussion

As simplified analogues of natural proteins, synthetic polypeptide polymers constitute important building blocks to develop highly functional materials. [25] Water-soluble polypeptides that adopt stable secondary structures are attractive because they pave the way to biomimetic structures that reproduce crucial features of natural peptides [22,26]. We recently developed thymidine-based nucleopolypeptides presenting stable β-sheets conformation in aqueous solution and exhibiting selective DNA-binding with simple sequences displaying adenine [21,22]. Herein, pursuing a macromolecular engineering design, we contemplated an introduction of such thymidine nucleobases onto multiblock polymeric backbone incorporating one clickable poly(γ-propargyl-l-glutamate) segment (Scheme 1).

The first step of our nucleopolypeptide design involved the synthesis of the polypeptide backbone [27,28]. Two copolymers with different architectures, diblock and triblock, were prepared in order to investigate the influence of the architecture on the morphology of nanoobjects formed by self-assembly in water. Both diblock and triblock copolymers, PBLG-*b*-PPLG and PPLG-*b*-PBLG-*b*-PPLG respectively, were achieved using a two-step ring-opening polymerization approach. First, macroinitiators were obtained by initiating the ring-opening polymerization of BLG-NCA in DMF at 0 °C, as a means to achieved controlled ROP with either hexylamine (PBLG) or ethylene diamine (PBLG-ED-PBLG) [29]. The polymerization reactions were monitored by FTIR following the disappearance of the BLG-NCA peaks (see Appendix A). As shown in Figure 1, upon purification, further analysis of these macroinitiators by ^1^H NMR analysis allowed to deduce number-average degrees of polymerization ($\bar{X_{n}}$) of 19 and 39 for PBLG and PBLG-ED-PBLG respectively.

Multiblock copolymers were further obtained by ring-opening polymerization of γ-propargyl-l-glutamate NCA (PLG-NCA) from macroinitiators PBLG and PBLG-ED-PBLG using a PLG-NCA/macroinitiator ratio of 25:1 in DMF and at RT. It should be noted that, the synthesis of the monomer PLG-NCA was achieved according to our previous report [21]. The polymerizations were monitored by FTIR following the disappearance of the PLG NCA peaks. Upon purification, analysis of the block copolymers was analyzed by size exclusion chromatography in DMF and ^1^H-NMR spectroscopy (Figure 2 and Appendix A), to check the good agreement between the polymer compositions with the monomers feed ratio. As summarized in Table 1, PBLG-*b*-PPLG diblock and PPLG-*b*-PBLG-*b*-PPLG triblock copolypeptides exhibit low dispersities (Ð < 1.2) and weight average molar masses ($\bar{Mn}$) of 5900 g/mol and 13,400 g/mol, respectively, close to those estimated from NMR analyses.

Functionalization of diblock and triblock copolypeptides with thymidine was achieved through CuAAC reaction in anhydrous and deoxygenated DMF using CuBr/PMDETA as catalytic system. Upon extensive purification by dialysis, the presence of the nucleobase in the final polymer structure was verified by ^1^H NMR and ^31^P NMR. As depicted in Figure 3, integration of the ^1^H NMR signals indicated a quantitative coupling (integration of the signal a compared to the signal i). This was further confirmed by the complete disappearance of the ^31^P NMR signal of the AZTP (Figure 3, right spectra) and of the N_3_ stretching on IR analyses (Appendix A).

Interestingly, both amphiphilic copolymers were spontaneously able to fully disperse in water, forming opalescent solutions, a qualitative indication for spontaneous self-assembly into nanoobjects. Indeed, thymidine-containing homopolypeptide have been showed to stabilize β-sheet structuring while keeping at the same time water solubility [22]. On another hand, it is well documented that poly(γ-benzyl-l-glutamate) block forms strong α-helices in aqueous solutions, a rod-like feature that was previously used to tune self-assembly in water with tree-like glycopolypeptides [16]. The presence of the secondary structures was confirmed by FTIR analyses (Appendix A).

To better understand this self-assembly, diblock (PBLG-*b*-PPLGNu) and triblock (PPLGNu-*b*-PBLG-*b*-PPLGNu) copolypeptides were directly solubilized in milliQ water at 2.5 mg/mL. The resulting nanoobject obtained were characterized by light scattering (LS) and transmission electron microscopy (TEM) without further purification. As shown in Figure 4, diblock copolypeptide self-assembled into nanoparticles with Z-average hydrodynamic radius (R_H_) equal to 14 nm, which is characteristic for spherical micelles [30]. This observation was consistent with TEM analysis (Figure 4A), where very small spherical micelles were observed. In marked contrast to previous similar design involving PBLG for which vesicles were formed [15], this first result indicated that the rod-induced packing of the hydrophobic segment was strongly counterbalanced by the highly charged thymidinylated segments that introduced strong electrostatic repulsion of the corona and consequently an increase of the interfacial curvature. In the case of triblock copolypeptides, dynamic light scattering (DLS) analysis revealed the formation of larger nanoobjects with an averaged R_H_ = 108 nm (see Figure 4B and Appendix A). A Z-average radius of gyration (R_g_) equal to 111 nm was determined by analyzing the static light scattering data using the Guinier plot method. A so-called ρ-ratio (ρ = R_g_/R_H_), which is an important indication of the morphology of scattering nanoparticles, close to 1 (ρ = 1.02) was obtained, suggesting the formation of vesicular morphology with the triblock [15,31]. These findings agreed well with the TEM image, where spherical hollow nanoobjects were clearly observed (Figure 4B and Appendix A). As compared to the diblock, the formation of vesicular structures with the triblock was assigned to the longer PBLG segments which was able to better stabilize a flat interface necessary for vesicle formation due to stronger rod-rod interaction between α-helices, that counter-balanced the electronic repulsion from the thymidinylated block. However, it is important to consider that the triblock could hypothetically be bent at the ethylene diamine further (1) inducing hydrophobic interaction between the two PBLG blocks, also (2) resulting in the possible formation of polymeric micelles with the cores of the bent PBLG blocks. This fact can partly explain the dispersity observed in Malvern analyses (Figure 4) and multiangle analyses confirmed the presence of smaller spherical objects (Appendix A). Overall, these results suggested a contradictory reciprocity between two phenomena—the alignment of the PBLG macrodipoles (the hydrophobic rod packing) and the negative charges repulsion of the ionic polypeptides. These two phenomena, in equilibrium, undoubtedly induced a form of dynamism in water that could explained the self-assembly observed upon direct solubilization in this solvent.

In a final step, to verify that the nucleobases present at the surface of the nanomaterials were available to mediate an efficient interaction with DNA, DNA binding assays with short single stranded oligoA DNA (20 mers) were performed. These assays involved mixing of the DNA solution (15 µM) with the two different self-assembled nanostructures (50 µg/mL) and check upon ultra-filtration the oligoA DNA retained (UV absorbance of adenine at 260 nm). As depicted on Figure 5, upon ultrafiltration using a MWCO of 30kDa, oligoA DNA was found retained with both nanomaterials. After centrifugation the eluted solution did not show any absorbance at 260 nm absorbance revealing that the oligoDNA stayed sequestered onto the microfilter upon association with the nanoassemblies (columns with (AAAAA)_4_ and NP in Figure 5). The control containing oligoDNA alone, when passed through the same cartridge under the same conditions, was not retained by the cartridge (left column, Figure 5). Overall, these results demonstrated that the thymidine in the shell of the nanomaterials were available to mediate an efficient interaction with a short oligo A sequence. In previous work, no *T*_m_ value was observed from hybridization experiments involving thymidinylated homopolypeptides [22]. Indeed, even if this interaction was previously found coarser than in regular base pairing occurring between two double stranded DNA, observing this same coarse interaction on the surface of nanoobjects opens up prospects for interesting surface functionalization or therapeutic targeting.

## 4. Conclusions

In summary, we have presented a simple macromolecular design to prepare amphiphilic nucleobase-containing polypeptides fully based on amino acid and nucleotide building blocks. Using sequential and controlled NCA polymerization and efficient click thymidinylation afforded unconventional amphiphilic architecture in which hydrophilic blocks structured in β-sheets are associated to a hydrophobic block structured in alpha helix. By varying the macromolecular engineering (diblock versus triblock), the morphology of the corresponding nanostructures was changed and afforded nanomaterials able to efficiently interact with oligo DNA sequences. From a materials perspective, the excellent control both in structure and in function drawn by our polymeric design hold promises to develop nanosized drug carriers for drug delivery systems targeting oligoA DNA sequences.

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
