# Peer review of "Amphiphilic Nucleobase-Containing Polypeptide Copolymers—Synthesis and Self-Assembly"

_polymers, 2020, doi:10.3390/polym12061357_

Round 1

Reviewer 1 Report

Bonduelle et al. developed a simple macromolecular design to prepare amphiphilic nucleobase-containing polymers. The manuscript is well-written, and the experimental results are summarized clearly. I recommend this manuscript can be published in Polymers after a minor revision. Followings are the remarks for the author:

  1. In Figure 4A, the author listed a PDI value of 0.31. However, a PDI value should be never lower than 1. The author should re-calculate this number.
  2. In addition to TEM analysis, AFM is suggested to be included to investigate the studied polymer particles since these particles have small dimension of 20~100nm.

Reviewer 2 Report

This paper describes the self-assembly of the nucleobase-polypeptide diblock and triblock copolymers.  While I think this study should attract the attention of chemists and make a contribution to fields of polymer chemistry and colloid chemistry, I have a reservation that should be addressed before publication.  My comments are the following.

  1. The vesicle formation by the triblock copolymer is doubtful because the triblock copolymer is not linear, unlike shown in Figure 4.  The copolymer has the potential to be bent at the ethylene diamine due to the hydrophobic interaction between the two PBLG blocks, resulting in the formation of large-size micelles with the cores of the bent PBLG blocks.  In fact, many objects much smaller than the 100-nm objects, are observed in Figure 4B.
  2. If the triblock copolymer is linear, the authors need to compare the length of the triblock copolymer with the wall thickness of the vesicles.

The authors should clarify the above problem and promote better discussion.

Reviewer 3 Report

The manuscript reports the synthesis and self-assembly of amphiphilic diblock and triblock copolypeptides. Overall, the results are lack of novelty and significance. The data quality is poor. I will not recommend to publish this work in Polymers.

The authors attempt to use many big or fancy words to decorate their commonplace results, like nanoobjects and macromolecular engineering approach, which I think are very inappropriate. The assembled structures are just regular micelles, and the approach used to self-assemble amphiphilic polymers into micelles has been reported for more than two decades tracing back to Eisenberg’s paper.  

The full name of CuAAC should be presented when it appears for the first time in main text.

It is confused to me how the self-assembly take places. In line 240, the authors claim aqueous dispersion without aggregates were obtained. Dispersion and self-assembly is a contradictory process.

Line 278, followed by FTIR. Please show FTIR results.

In the main text, a SEC profile incorporating four polymers in Table 1 is required.

The quality of HNMR data showing in Figure 3 is not satisfactory. They have to be improved.

Since amphiphilic copolymers spontaneously form assembled structures in water, D2O should not be used as a solvent for NMR measurement.

The quality of TEM images are very poor, and they have to be improved. From the current TEM images, one can see many unassembled polymers and other impurities.

For Figure 4B, please show the same type of DLS result of Figure 4A, which can provide a more straightforward impression on the size and distribution of vesicles.

The cartoons of spherical micelles and vesicles in Figure 4 are misleading. It looks like further phase separation take places between blue and red polymer. However, the current data is impossible to prove such assumed structures.  

Figure 5, parallel experiments have to be performed and an error bar is needed.

Round 2

Reviewer 2 Report

The revised manuscript is acceptable for publication in Polymers.

Author Response

Extensive proofreading has been achieved to correct typos and English mistakes.